# Adult Botulism of Unknown Source with Post-Toxin Anti-GQ1b Antibodies: Implications for Molecular Mimicry—A Case Report

**DOI:** 10.3390/neurolint18010008

**Published:** 2025-12-29

**Authors:** Regev Cohen, Adi Hersalis Eldar, Yaron River, Ofir Schuster, Zina Baider, Shelly Lipman-Arens, Yael Galnoor Tene, Linor Ishay, Lamis Mahamid, Olga Feld Simon, Nina Avshovitch, Alvira Zbiger, Eran Diamant, Amram Torgeman, Elad Milrot, Ofir Israeli, Shlomo Shmaya, Itzhak Braverman, Shlomo E. Blum

**Affiliations:** 1Infectious Diseases Unit, Hillel Yaffe Medical Center, Hadera 38100, Israel; shellyl@hymc.gov.il (S.L.-A.); yaelgal@hymc.gov.il (Y.G.T.);; 2Rappaport Faculty of Medicine, Technion-Israel Institute of Technology, Haifa 3200003, Israel; 3Neurology Department, Hillel Yaffe Medical Center, Hadera 38100, Israel; adie@hymc.gov.il (A.H.E.); yaronr@hymc.gov.il (Y.R.); 4Department of Infectious Diseases, Israel Institute for Biological Research, Ness Ziona 74100, Israel; ofirsc@iibr.gov.il (O.S.); eladm@iibr.gov.il (E.M.); shlomosh@iibr.gov.il (S.S.); 5Botulism National Reference Laboratory, Kimron Veterinary Institute, Bet Dagan 5025001, Israel; zinab@moag.gov.il (Z.B.);; 6Internal Medicine D, Hillel Yaffe Medical Center, Hadera 38100, Israel; ninaa@hymc.gov.il (N.A.); 7Microbiology Laboratory, Hillel Yaffe Medical Center, Hadera 38100, Israel; 8Ear Nose and Throat Department, Hillel Yaffe Medical Center, Hadera 38100, Israel

**Keywords:** botulism, anti GQ1b, autoimmune, molecular mimicry, Miller–Fisher syndrome, case report

## Abstract

**Background**: Botulism is a rare but potentially fatal neuroparalytic illness caused by *Clostridium botulinum* neurotoxins (BoNTs). While adult cases usually result from foodborne exposure or wound infection, intestinal colonization is exceedingly uncommon. Diagnosis can be delayed by overlap with other neuromuscular syndromes, and confirmation requires specialized assays. Anti-GQ1b antibodies, classically associated with Miller–Fisher syndrome (MFS), have rarely been reported in confirmed botulism, raising questions about shared pathophysiology. **Case Presentation**: We describe an adult patient with acute dyspnea, xerostomia, and cranial neuropathies. No foodborne source was identified, but intestinal colonization of BoNT/B toxigenic *Clostridium botulinum* was confirmed by stool enrichment and mouse lethality bioassay. The patient improved promptly following heptavalent antitoxin. Unexpectedly, anti-GQ1b antibodies were detected during recovery, a finding typically linked to MFS rather than botulism. **Discussion**: This case illustrates the diagnostic challenges of sporadic cases of botulism, especially when respiratory compromise and autonomic dysfunction dominate the initial presentation. The autoantibodies finding raises the possibility of molecular mimicry, whereby toxin–ganglioside interactions expose neuronal epitopes and trigger an immune response. While causality cannot be proven, the overlap between botulism and GQ1b-positive neuropathies merits further investigation. **Conclusions**: Clinicians should maintain high suspicion for botulism in adults with acute dyspnea, especially when associated with cranial neuropathies, even in the absence of foodborne exposure. Anti-ganglioside antibodies in this context should be interpreted with caution, as they do not exclude botulism but may highlight immunological overlap with autoimmune neuropathies.

## 1. Introduction

Botulism is a rare, life-threatening neuroparalytic disorder caused by botulinum neurotoxins (BoNTs) produced by *Clostridium botulinum*. While most adult cases are foodborne or wound-related, intestinal colonization with in vivo toxin production is exceptionally uncommon outside infancy. Diagnosis is challenging because confirmatory testing is available only in specialized laboratories and may yield false-negative results early in the course. Prompt clinical recognition is essential, as outcomes depend on rapid administration of botulinum antitoxin.

Anti-ganglioside antibodies, particularly anti-GQ1b, are strongly associated with Miller–Fisher syndrome (MFS) and related Guillain–Barré Syndrome (GBS) spectrum disorders. Their presence in confirmed botulism cases is exceedingly rare but raises the possibility of molecular mimicry between ganglioside-binding toxins and host neuronal membranes. Here we describe an adult patient with laboratory-confirmed BoNT/B intoxication of unknown source who developed anti-GQ1b antibodies during recovery. This case highlights both the diagnostic challenges of botulism and the potential immunological overlap with autoimmune neuropathies.

## 2. Case Report

A 37-year-old male presented to the emergency department with acute dyspnea and generalized weakness. His medical history included obesity, type 2 diabetes and psoriasis. As an exterminator, he had regular exposure to various insecticides. His medications included insulin and omeprazole. His mother had a history of seropositive Myasthenia Gravis (MG).

He was afebrile, normotensive and dyspneic, with an oxygen saturation of 91% while breathing ambient air. Electrocardiograph showed signs of S wave in lead I, Q and inverted T waves in lead III, but chest computerized tomography (CT) angiography ruled out pulmonary embolism. Laboratory tests revealed mild thrombocytopenia (127,000 cells/µL) and hyponatremia (133 mEq/L). C-reactive protein was 6 mg/L (normal < 5 mg/L).

On hospital day (HD) 1, he reported persistent dyspnea and difficulty swallowing. An ear, nose and throat examination revealed dry mouth and acute nasopharyngitis. Treatment with dexamethasone and intravenous Augmentin (ampicillin-clavulanate) was initiated. A toxicology consultation did not identify any specific toxidrome. On HD2, he developed slurred speech and diplopia in upward gaze. By HD3, he experienced mouth and tongue numbness and required oxygen support. Suspicion of nuchal rigidity led to a lumbar puncture, which yielded normal cerebrospinal fluid findings. Repeated pharyngeal examination revealed brown crust and acute nasopharyngeal inflammation. By HD4, the patient exhibited dysarthria, difficulty swallowing fluids with drooling, nasal dysphonia, bilateral ptosis, diplopia in all directions and severe constipation. On neurological examination, pupils were normal and reactive. Left sternocleidomastoid weakness and proximal muscle weakness (3/5) in both upper and lower limbs were evident. Sensation and deep tendon reflexes were normal, and no pathological reflexes or cerebellar signs were seen. Appendix A depicts the patient attempting tongue protrusion, with evident strabismus and ptosis. In Video Clip S1, the patient demonstrates difficulty moving his tongue in response to verbal instructions.

The differential diagnosis included MG and variants of GBS such as MFS and Pharyngeal-Cervical-Brachial (PCB) syndrome. An urgent electromyography and nerve conduction studies (EMG/NCS) revealed small compound muscle action potentials (CMAPs) with normal latency and conduction velocity and without conduction blocks. Slow frequency (3 Hz) repetitive nerve stimulation was normal, but high frequency (50 Hz) and stimulation after short isometric exercise showed incremental response, findings consistent with presynaptic neuromuscular disorder. Since Lambert–Eaton Myasthenic Syndrome (LEMS) was unlikely given the rapid onset and deterioration, botulism was suspected.

The patient was transferred to the intensive care unit (ICU) and administered equine-derived heptavalent botulinum antitoxin (HBAT). Fecal samples were sent to the Israel Institute for Biological Research (IIBR). The patient showed rapid improvement following antitoxin administration. His dyspnea improved within a day, and over the next two days, his ptosis, diplopia, tongue movement, neck strength, and deltoid muscle function improved as well. He was discharged two days after receiving the antitoxin.

Three weeks post-discharge, the patient continued to experience exercise dyspnea and severe autonomic dysfunction, including dry mouth and hands, orthostatic hypotension, anhidrosis, impotence and constipation. He had lost 10 Kg due to swallowing difficulty. All symptoms resolved after 4 months.

A direct Polymerase Chain Reaction (PCR) test for C. botulinum types A and B from feces yielded negative results. However, anaerobic enrichment of the fecal samples enabled the detection of botulinum neurotoxin B (BoNT/B) using a mouse lethality bioassay (MLB) at the National Research Laboratory at the Kimron Veterinary Institute, which was subsequently confirmed using genetic and immunological assays at the IIBR. The Ministry of Health was notified, but the source of infection remained unidentified. Autoantibodies obtained on HD5 returned positive for anti-GQ1b antibodies.

## 3. Discussion

This case documents a classical presentation of adult botulism caused by BoNT/B, confirmed through stool enrichment and MLB, despite the absence of an identifiable source. The patient’s rapid recovery after administration of heptavalent antitoxin highlights the importance of early recognition and intervention, consistent with current guidelines [1]. The initial symptom of acute dyspnea without prominent neurological findings delayed the diagnosis, a pattern that has been repeatedly observed. In a review of 402 confirmed foodborne and wound botulism cases studied from 1932–2015, 42% of patients presented with respiratory distress at admission, often before the development of weakness [2]. More recent CDC surveillance data similarly note that shortness of breath is documented in most patients within the first 48 h, nearly half of whom eventually required ventilation [1]. These data emphasize that respiratory compromise may dominate the early picture and should raise suspicion for botulism when combined with cranial involvement.

In this patient, xerostomia and nasopharyngeal inflammation further confounded the diagnosis, leading to inappropriate antimicrobial therapy. Such antibiotics may worsen the course by lysing vegetative *Clostridium botulinum* in the gut and releasing toxin. While this mechanism is best described in infant botulism, it may also be relevant in adults under similar circumstances [3]. The use of omeprazole, a proton pump inhibitor, likely altered gut flora and facilitated intestinal colonization [4,5]. These factors, combined with occupational exposure, make intestinal toxin production the most plausible infection route. This interpretation aligns with a concurrent Israeli infant case caused by BoNT/B without a foodborne source [6], suggesting an environmental reservoir.

Our patient had a rapid clinical response to HBAT administration, with dyspnea improving within hours, allowing him to be discharged from the ICU to the neurology department after one day, and home after two days. Data from 249 HBAT treatments showed symptom improvement started on a median of 2.4 days after administration, with benefits limited to stopping botulism progression rather than accelerating recovery [7]. This accelerated improvement may be attributable to the presence of BoNT/B, given that, in the referenced study, 74% of cases were associated with BoNT/A.

Laboratory confirmation of botulism is complex, and treatment should be provided based on clinical suspicion, as results may take days and can be falsely negative even in confirmed cases. The MLB, while labor-intensive and available only in specialized laboratories, remains the only Food and Drug Administration (FDA)approved test for detecting biologically active botulinum toxin in serum, stool, or gastric samples [1]. Our patient’s stool direct PCR test was negative. Final diagnosis was achieved only via anaerobic enrichment of stool, followed by MLB and PCR or immunoassays, illustrating the essential role of traditional confirmatory methods, with an emphasis on PCR from enriched stool culture.

A particularly intriguing feature of this case was the detection of anti-GQ1b antibodies after recovery, making this likely the second laboratory-confirmed botulism case reported with this finding [8]. The former published case was a 70-year-old male admitted with descending weakness, dysphagia, ataxia, ophthalmoplegia, areflexia, ptosis and mydriasis. He was treated with intravenous immunoglobulins (IVIG) for the diagnosis of MFS, but also with botulinum antitoxin and plasma exchange. High levels of anti-GQ1b antibodies were detected, and botulinum toxin was found in his canned food, though evidences of *Clostridium botulinum* could not be found in his serum or feces. Two additional case reports described clinical features that were consistent with botulism, developing shortly after the recent consumption of suspected foods; however, the diagnosis was revised to MFS when anti-GQ1b antibodies were detected, while botulism diagnosis could not be confirmed [9,10]. The concurrence of two rare diseases is intriguing and may not be coincidental.

Anti-GQ1b antibodies are typically associated with MFS and related Guillain–Barré spectrum disorders, where they target GQ1b gangliosides that are enriched in cranial nerves III, IV, and VI [11,12]. Clinical overlap between botulism and MFS is well recognized, particularly in ophthalmoplegia and autonomic dysfunction, raising the possibility of shared molecular targets.

In typical botulinum food intoxication, antibodies against the toxin are generally not produced, as the toxin acts rapidly and at very low doses in localized sites, before a significant systemic immune response can develop (“the toxic dose is less than the immunogenic dose”). But, the situation is different in cases of intestinal botulism, where endogenous antibody production was indeed demonstrated in both infants [5] and adults with bowel or microbiome dysfunction [13,14].

Botulinum neurotoxins bind presynaptic membranes via a dual-receptor mechanism that combines polysialogangliosides with protein receptors like synaptotagmin [15]. The ganglioside receptors for BoNT/A include GT1b and GD1a, while GQ1b is considered to have lower affinity, although Takamizawa et al. studied the receptor structure of BoNT/A using thin-layer chromatographic immunostaining and found GQ1b to be the most potent receptor, with lower binding to GT1b and GD1a gangliosides [16]. The ganglioside receptors for BoNT/B include GT1b and GD1a, but not GQ1b. It should be noted that GQ1b and GT1b are structurally very similar gangliosides, sharing the same terminal disialylated structure (disialylgalactose), which is the immunodominant epitope recognized by autoantibodies in GBS/MFS [17]. As GT1b and GQ1b differ only by the addition of one sialic acid residue, cross-reactivity between anti-GQ1b and anti-GT1a antibodies, both biomarkers of MFS, supports the concept of immunological overlap [18]. Several studies have shown that anti-GQ1b antibodies cross-react with GT1b by recognizing the shared terminal disialosyl motif common to several b-series gangliosides [19,20,21,22]. A study of patients with MFS or GBS with ophthalmoplegia found anti-GQ1b antibodies in all cases, and the same antibodies also reacted with GT1a gangliosides [23].

Theoretically, toxin–ganglioside binding might alter ganglioside conformation or expose hidden epitopes, triggering an autoimmune response through molecular mimicry. Continuous production of BoNT/B, as in cases of intestinal botulism and its binding to GT1b, especially if for prolonged periods of time, may result in an immune reaction and in the production of anti-GQ1b (and probably also anti-GT1b), given the ganglioside resemblance and antibodies’ indiscrimination between the two gangliosides.

The heavy chain of BoNT forms a channel within endosomes to translocate the light chain into the cytosol, a process normally shielded from immune surveillance [24]. However, experimental work suggests that toxin internalization may not always be immediate, and under certain conditions, intact toxin can recycle back to the synaptic surface [25]. Such persistence could provide another opportunity for immune recognition, although this remains speculative and unproven.

While our patient demonstrated both confirmed BoNT/B intoxication and anti-GQ1b positivity, causality cannot be established. The antibodies may have pre-existed, or they may represent a secondary epiphenomenon of intoxication. Unfortunately, a baseline pre-intoxication serum sample for this patient was unavailable. The coexistence of these two rare phenomena raises important questions about whether subclinical botulism could underlie some presentations currently attributed to idiopathic MFS variants, such as Bickerstaff brainstem encephalitis (BBE) or PCB weakness. Longitudinal studies measuring anti-ganglioside antibodies before and after botulism, coupled with structural analyses of ganglioside conformation post-toxin binding, will be critical to address this possibility. The possibility of intestinal infection with bacteria as *Campylobacter jejuni*, that are typically related to GBS or MFS, was considered unlikely, as the patient did not have diarrheal syndrome or fever in the preceding weeks before presentation, and *C. botulinum* was preset in the feces; although this option of co-infection was not tested nor excluded entirely. It could be argued that administering HBAT might affect the detection of anti-GQ1b antibodies; however, since the product consists only of Fc-depleted, antigen-binding fragment (FAB) equine antibodies, laboratory tests like Enzyme-Linked ImmunoSorbent Assay (ELISA) would not have mistakenly identified these HBAT fragments as autoantibodies.

The concurrent infant botulism case supports the likelihood of environmental exposure and highlights the need for improved surveillance of intestinal colonization in adults. As in our case, the clinical improvement of this infant was brisk [6]. Although clinical improvement after HBAT is anticipated, a very rapid improvement should not lead clinicians to reconsider botulism diagnosis solely on that basis. Finally, clinicians should recall that stool enrichment with confirmatory bioassays remains vital when initial PCR results are negative.

## 4. Conclusions

This case highlights the importance of considering botulism in adults presenting with acute dyspnea, even in the absence of cranial neuropathies or a clear foodborne source. Stool enrichment and functional bioassays remain essential when rapid diagnostics are inconclusive. Considering the limitation posed by the absence of a molecular model for the proposed theorem, the unexpected detection of anti-GQ1b antibodies raises the possibility that toxin–ganglioside interactions may trigger autoimmune phenomena in predisposed individuals, or that subclinical intestinal botulism could contribute to antibody-positive presentations currently attributed to MFS variants. Although causality cannot be established, this overlap underscores the need for cautious interpretation of anti-ganglioside serology in suspected botulism and for further research into the immunological consequences of toxin binding.

## Data Availability

Data related to this case report are available upon request.

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
