# Peer review of "Adult Botulism of Unknown Source with Post-Toxin Anti-GQ1b Antibodies: Implications for Molecular Mimicry—A Case Report"

_2035-8377, 2025, doi:10.3390/neurolint18010008_

Round 1
Reviewer 1 Report
Comments and Suggestions for Authors
This case report details a patient with botulism type B infection that also presented with Miller-Fisher syndrome (MFS), a rare variant of Guillain-Barré syndrome that was confirmed by the presence of autoantibodies towards ganglioside GQ1b. This is the second confirmed case of coincident botulism and MFS but differs from the previous report in that a B serotype botulinum toxin (BoNT)-producing strain is identified and the presence of intestinal infection is shown. This study is important because several previous suspected cases of botulism have been reassigned to MFS, but the authors point out that the presence of anti-GQ1b antibodies does not rule out botulism comorbidity.
It is known that BoNT/B binds to gangliosides, and although it does not interact strongly with GQ1b the authors speculate that the toxin interaction with other gangliosides might create epitopes that mimic the GQ1b structure. The authors admit that this is speculation without any supporting evidence. Moreover, the authors should comment on the low likelihood that a toxin-ganglioside complex could generate an immune response sufficient to generate clinically relevant anti-GQ1b autoantibodies without also the toxin alone generating a neutralising response.
An alternative possibility worthy of comment is co-infection with a gram-negative bacterium in addition to clostridium botulinum, especially considering the apparent gut immune system deficiency of the patient with altered gut flora due to medication already highlighted as a factor that increases likelihood of infection. For example, campylobacter jejuni is a common infection that has been implicated in MFS due to the structure of its lipooligosaccharides mimicking gangliosides (Yuki, N., 1997, J. Infect. Dis. doi: 10.1086/513800).
Not all cited references are in the bibliography, and some in-text citations are not in the correct number format.
Not all abbreviations are included in the list at the end of the report. Some abbreviations are not defined (e.g. BBE).
Line 190-191 contradicts line 180-181. At line 197, should ‘antitoxin’ be replaced with ‘intravenous immunoglobulin’?
Author Response
Thank you so much for taking the time to review this manuscript. Please find the detailed responses below and the corresponding revisions/corrections highlighted/in track changes in the re-submitted files.
Reviewer 1
This case report details a patient with botulism type B infection that also presented with Miller-Fisher syndrome (MFS), a rare variant of Guillain-Barré syndrome that was confirmed by the presence of autoantibodies towards ganglioside GQ1b. This is the second confirmed case of coincident botulism and MFS but differs from the previous report in that a B serotype botulinum toxin (BoNT)-producing strain is identified and the presence of intestinal infection is shown. This study is important because several previous suspected cases of botulism have been reassigned to MFS, but the authors point out that the presence of anti-GQ1b antibodies does not rule out botulism comorbidity.
It is known that BoNT/B binds to gangliosides, and although it does not interact strongly with GQ1b the authors speculate that the toxin interaction with other gangliosides might create epitopes that mimic the GQ1b structure. The authors admit that this is speculation without any supporting evidence.
Moreover, the authors should comment on the low likelihood that a toxin-ganglioside complex could generate an immune response sufficient to generate clinically relevant anti-GQ1b autoantibodies without also the toxin alone generating a neutralising response.
We thank the reviewer for this important subject.
We added a paragraph regarding this issue. In cases of botulism intoxication, the immune response is typically incapable of producing neutralizing antibodies given their low dose and rapid action; but in cases of intestinal botulism, where the toxin is produced in situ and the colonization may be protracted, endogenous antibody production was indeed demonstrated. We brought several references for both infants and adults that presented anti-toxin production. We added this section to the text:
“In typical botulinum food intoxication, antibodies against the toxin are generally not produced, as the toxin acts rapidly and at very low doses in localized sites, before a significant systemic immune response can develop (“the toxic dose is less than the immunogenic dose”). But, the situation is different in cases of intestinal botulism, where endogenous antibody production was indeed demonstrated in both infants (5) and adults with bowel or microbiome dysfunction (16, 17).”
An alternative possibility worthy of comment is co-infection with a gram-negative bacterium in addition to clostridium botulinum, especially considering the apparent gut immune system deficiency of the patient with altered gut flora due to medication already highlighted as a factor that increases likelihood of infection. For example, campylobacter jejuni is a common infection that has been implicated in MFS due to the structure of its lipooligosaccharides mimicking gangliosides (Yuki, N., 1997, J. Infect. Dis. doi: 10.1086/513800).
We do not think our patient had concomitant/preceding infection with C. jejuni, as he did not recall fever or diarrhea and we added this information to the text:
“The possibility of intestinal infection with bacteria as Campylobacter jejuni, that are typically related to GBS or MFS, was considered unlikely, as the patient did not have diarrheal syndrome or fever in the preceding weeks before presentation, and C. botulinum was preset in the feces.”
Not all cited references are in the bibliography, and some in-text citations are not in the correct number format.
We thank the reviewer for noting this, and we corrected the references format and numbering.
Not all abbreviations are included in the list at the end of the report. Some abbreviations are not defined (e.g. BBE).
We added all the missing abbreviations, including those that were defined in the main text.
Line 190-191 contradicts line 180-181.
We understand the confusion that was caused by these two sentences. Our intention was to focus the reader on the rapidity of the response to anti-toxin, as the improvement in our case occurred within days (and was expected to occur at a more gradual pace). We chose to delete the sentence from lines 180-181, but we added a paragraph to the discussion to emphasize this point:
“Our patient had a rapid clinical response to HBAT administration, with dyspnea improving within hours, allowing him to be discharged from the ICU to the neurology department after one day, and home after two days. Data from 249 HBAT treatments showed symptom improvement started on a median of 2.4 days after administration, with benefits limited to stopping botulism progression rather than accelerating recovery (7). This accelerated improvement may be attributable to the presence of BoNT/B, given that, in the referenced study, 74% of cases were associated with BoNT/A”.
At line 197, should ‘antitoxin’ be replaced with ‘intravenous immunoglobulin’?
We believe that the reviewer meant line 191 and not 197, and to the idea that IVIG given to treat MFS or GBS with positive clinical response, should not deter the physician to still suspect botulism. But what we meant relates to the rapidity of the response to HBAT – hence we added the paragraph cited in the previous remark, and we believe that the text in much clearer now. We also changed the sentence in lines 191 to be:
“Clinicians should not dismiss botulism diagnosis based on a very rapid improvement after HBAT administration…”
Reviewer 2 Report
Comments and Suggestions for Authors
The abstract is well structured and presents a clinically important case alongside an intriguing pathophysiological discovery. The introduction is concise and informative. It clearly highlights the rarity of Clostridium colonisation of the intestine in adults and the associated diagnostic challenges. The case report is presented in clear chronological order, with key medical history data provided, and the clinical approach to differential diagnosis and the difficulties encountered during the diagnostic phase are clearly demonstrated. The discussion goes beyond a simple description of the clinical case, offering a substantive scientific analysis. The conclusions accurately reflect the case's key points.
It is recommended that the content of Figures S1 and S1 is briefly described in the text, as the case description itself does not indicate what can be seen in them.
Questions:
1. Is there any information about a family history of myasthenia gravis and its potential role in this case?
2. It is stated that antibodies were detected on the fifth day of hospitalisation (HD5). However, it would be useful to clarify how this relates to the administration of antitoxin.
Author Response
The abstract is well structured and presents a clinically important case alongside an intriguing pathophysiological discovery. The introduction is concise and informative. It clearly highlights the rarity of Clostridium colonisation of the intestine in adults and the associated diagnostic challenges. The case report is presented in clear chronological order, with key medical history data provided, and the clinical approach to differential diagnosis and the difficulties encountered during the diagnostic phase are clearly demonstrated. The discussion goes beyond a simple description of the clinical case, offering a substantive scientific analysis. The conclusions accurately reflect the case's key points.
We thank the reviewer for this comment.
It is recommended that the content of Figures S1 and S1 is briefly described in the text, as the case description itself does not indicate what can be seen in them.
We added this paragraph to the text:
“Figure S1 depicts the patient attempting tongue protrusion, with evident strabismus and ptosis. In Video Clip S1, the patient demonstrates difficulty moving his tongue in response to verbal instructions.”
Questions:
- Is there any information about a family history of myasthenia gravis and its potential role in this case?
The patient’s mother had MG. We may speculate on familial tendency for autoimmune diseases, but we do not see a way to integrate this speculation into the text.
- It is stated that antibodies were detected on the fifth day of hospitalisation (HD5). However, it would be useful to clarify how this relates to the administration of antitoxin.
It could be argued that administering HBAT might affect the detection of anti-GQ1b antibodies; however, since the product consists only of Fc-depleted, FAB equine antibodies, laboratory tests like ELISA would not have mistakenly identified these HBAT fragments as autoantibodies. We added this to the text.
Reviewer 3 Report
Comments and Suggestions for Authors
The sole reason for rejection: The study lacks molecular modeling results for the GQ1b antibody, which is a major source of credibility for the research.
Author Response
Reviewer 3 –
The sole reason for rejection: The study lacks molecular modeling results for the GQ1b antibody, which is a major source of credibility for the research.
The association between anti-GQ1b antibodies and intestinal botulism has not been investigated; therefore, an established molecular model is currently unavailable. We expanded our discussion to provide literature evidence of endogenous antibody production in cases of intestinal botulism (in adults and infants) and on the molecular similarity between GQ1b and GT1b and antibody indiscrimination between the two type-b gangliosides (the similarity between the two gangliosides can be seen in the figure that is included in the end of this document). We suggest that production of botulinum toxin B in cases of chronic gut colonization may interact with GT1b gangliosides and trigger an autoimmune response, producing antibodies that may interact with GT1b and GQ1b gangliosides, resulting in their detection.
Although we cannot prove this theorem using a molecular model, we believe that the association between these two rare diseases (intestinal botulism and MFS) merits further exploration, hence the importance of publishing this association and idea.
The discussion was expanded with these explanations and references.
We hope we have addressed the reviewer’s concerns.
Round 2
Reviewer 1 Report
Comments and Suggestions for Authors
The authors are thanked kindly for their consideration of all the points raised in my report. The point is well made that the authors had no reason to suspect a second bacterial coinfection due to an absence of fever or diarrhoea. However, the authors themselves speculate that a prolonged subclinical infection with Clostridium botulinum might have been able to produce an immune response to a toxin-bound ganglioside complex. That the patient was infected with Clostridium botulinum is not disputed; the authors detected a BoNT/B producing strain in a culture elaborated from the patient’s faeces, using a sensitive and specific PCR assay. However, this assay does not rule out the possibility of a coinfection to which the same ‘subclinical yet immunogenic’ argument could be postulated. Such a coincident infection was not tested for, so absence cannot be excluded. Furthermore, as the authors correctly highlight, intestinal infection with Clostridium botulinum is most likely in cases with microbiome dysfunction (L187 in version 2 with changes displayed) typified by altered gut flora (L141). Therefore, this reviewer remains unconvinced by the authors dismissal of the possibility of coinfection with a bacteria producing an immune response cross-reactive with neural gangliosides. By their own admission, causality cannot be established between BoNT/B intoxication and the production of anti-GQ1b antibodies (L219), and presence of the antibodies may have preceded BoNT intoxication (L220). Thus, the sections exploring the possible causes of anti-GQ1b production should be more balanced regarding the possible source of the antigen responsible.
Thanks for corrections to reference lists, citations, and abbreviations list. Please add definitions for CT and HD to the abbreviations list and define IVIG in the text (L168).
Thanks for the clarification regarding the fast recovery after HBAT administration. I still have difficulty with the statement at L239 “Clinicians should not dismiss botulism diagnosis based on a very rapid improvement after antitoxin HBAT administration” because improvement after HBAT administration is expected for botulism even if the recovery was faster than anticipated. The fast recovery is interesting because it suggests that, in this case at least, that 1. the persistence of botulism symptoms required a source of BoNT that was accessible to injected HBAT, 2. when this source was neutralised intoxicated nerve terminals were able to recover quickly and 3. the BoNT/B protease involved may be relatively unstable.
Author Response
We thank the reviewer for the constructive comments.
We will adress them one by one:
Comment 1 - "...Thus, the sections exploring the possible causes of anti-GQ1b production should be more balanced regarding the possible source of the antigen responsible".
As we could not exclude this option, we added to the text the option of co-infection causing autoimmune phenomena. The text now reads:
The possibility of intestinal infection with bacteria as Campylobacter jejuni, that are typically related to GBS or MFS, was considered unlikely, as the patient did not have diarrheal syndrome or fever in the preceding weeks before presentation, and C. botulinum was preset in the feces; although this option of co-infection was not tested nor excluded entirely."
Comment 2 - "Please add definitions for CT and HD to the abbreviations list and define IVIG in the text (L168)."
We added the two abbreviations and defined IVIG as requested.
Comment 3 - "...I still have difficulty with the statement at L239 “Clinicians should not dismiss botulism diagnosis based on a very rapid improvement after antitoxin HBAT administration” because improvement after HBAT administration is expected for botulism even if the recovery was faster than anticipated..."
We changed the text to a more subtle version and it now reads: "Although clinical improvement after HBAT is anticipated, a very rapid improvement should not lead clinicians to reconsider botulism diagnosis solely on that basis..."
Reviewer 2 Report
Comments and Suggestions for Authors
Manuscript ready to publish
Author Response
Thanks
Round 3
Reviewer 1 Report
Comments and Suggestions for Authors
Cite a paper that identified a link between MFS and a bacterial infection.
e.g. Koga, M. et al., 2005, Neurology 64, 1605-1611